# Evaluation of Regional Air Quality Models over Sydney, Australia: Part 2, Comparison of PM$_{2.5}$ and Ozone

**Elise-Andrée Guérette** [1,2,3,*], **Lisa Tzu-Chi Chang** [4], **Martin E. Cope** [3], **Hiep N. Duc** [4], **Kathryn M. Emmerson** [3], **Khalia Monk** [1,4], **Peter J. Rayner** [5], **Yvonne Scorgie** [4], **Jeremy D. Silver** [5], **Jack Simmons** [1,2], **Toan Trieu** [4], **Steven R. Utembe** [5,6], **Yang Zhang** [7,8] and **Clare Paton-Walsh** [1,2,*]

1   Centre for Atmospheric Chemistry, University of Wollongong, NSW 2522, Australia;
    Khalia.Monk@environment.nsw.gov.au (K.M.); js828@uowmail.edu.au (J.S.)
2   School of Earth, Atmospheric and Life Sciences (SEALS), University of Wollongong, NSW 2522, Australia
3   Commonwealth Scientific and Industrial Research Organization (CSIRO), Oceans and Atmosphere,
    Aspendale, VIC 3195, Australia; martin.cope@csiro.au (M.E.C.); kathryn.emmerson@csiro.au (K.M.E.)
4   New South Wales Department of Planning, Industry and Environment, Lidcombe, Sydney, NSW 2141,
    Australia; LisaTzu-Chi.Chang@environment.nsw.gov.au (L.T.-C.C.);
    Hiep.Duc@environment.nsw.gov.au (H.N.D.); Yvonne.Scorgie@environment.nsw.gov.au (Y.S.);
    toan.trieu@environment.nsw.gov.au (T.T.)
5   School of Earth Sciences, University of Melbourne, VIC 3010, Australia; prayner@unimelb.edu.au (P.J.R.);
    jeremy.silver@unimelb.edu.au (J.D.S.); steven.utembe@epa.vic.gov.au (S.R.U.)
6   Environmental Protection Agency (EPA), VIC 3085, Australia
7   Department of Civil and Environmental Engineering, Northeastern University (NU), Boston, MA 02115,
    USA; ya.zhang@northeastern.edu
8   Department of Marine, Earth and Atmospheric Sciences, North Carolina State University (NCSU), Raleigh,
    NC 27695, USA
*   Correspondence: elise-andree.guerette@csiro.au (E.-A.G.); clarem@uow.edu.au (C.P.-W.)

**Abstract:** Accurate air quality modelling is an essential tool, both for strategic assessment (regulation development for emission controls) and for short-term forecasting (enabling warnings to be issued to protect vulnerable members of society when the pollution levels are predicted to be high). Model intercomparison studies are a valuable support to this work, being useful for identifying any issues with air quality models, and benchmarking their performance against international standards, thereby increasing confidence in their predictions. This paper presents the results of a comparison study of six chemical transport models which have been used to simulate short-term hourly to 24 hourly concentrations of fine particulate matter less than and equal to 2.5 μm in diameter (PM$_{2.5}$) and ozone (O$_3$) for Sydney, Australia. Model performance was evaluated by comparison to air quality measurements made at 16 locations for O$_3$ and 5 locations for PM$_{2.5}$, during three time periods that coincided with major atmospheric composition measurement campaigns in the region. These major campaigns included daytime measurements of PM$_{2.5}$ composition, and so model performance for particulate sulfate (SO$_4$$^{2-}$), nitrate (NO$_3$$^-$), ammonium (NH$_4$$^+$) and elemental carbon (EC) was evaluated at one site per modelling period. Domain-wide performance of the models for hourly O$_3$ was good, with models meeting benchmark criteria and reproducing the observed O$_3$ production regime (based on the O$_3$/NO$_x$ indicator) at 80% or more of the sites. Nevertheless, model performance was worse at high (and low) O$_3$ percentiles. Domain-wide model performance for 24 h average PM$_{2.5}$ was more variable, with a general tendency for the models to under-predict PM$_{2.5}$ concentrations during the summer and over-predict PM$_{2.5}$ concentrations in the autumn. The modelling intercomparison exercise has led to improvements in the implementation of these models for Sydney and has increased confidence in their skill at reproducing observed atmospheric composition.

**Keywords:** air quality modelling; model evaluation; $PM_{2.5}$; $O_3$

## 1. Introduction

Air quality models are used by government authorities to undertake both short-term, and strategic air quality forecasts. In support of this, scientists strive to improve the understanding of emissions and chemical and physical processes in the atmosphere that influence the composition of the air that we breathe [1]. In the region around Sydney, Australia, the two main atmospheric pollutants of concern are ozone ($O_3$) and particles with a diameter $\leq 2.5$ μm ($PM_{2.5}$) [2]. Air pollution is typically worse in western Sydney [3] and may be further exacerbated by expected large population growth in the next few years. It then follows that an improved understanding of the formation regimes for these two pollutants is paramount to developing effective mitigation policies [4].

There have been significant recent research efforts undertaken to gather observational databases, with the goal of improving photochemical $O_3$ and $PM_{2.5}$ modelling for air quality applications in Australia [5–7]. This has included modelling the air quality impacts from bushfires [8–10] and estimates of the health benefits to be gained by air quality improvements [11]. Modelling comparison exercises are an excellent way to assess the performance of air quality models and highlight any issues with the implementation of the different models [12–14]. Such comparisons can be used to evaluate the models, determine the accuracy of their predictions and ultimately build confidence in their performance, as demonstrated in several recent model intercomparisons such as the Air Quality Model Evaluation International Initiative (AQMEII) conducted in North America and Europe [14–18]. However, model intercomparison exercises involve a large amount of effort, and are very time consuming, and thus such an intercomparison study of hourly air quality models has not previously been undertaken in the Sydney region.

To address this gap, the Clean Air and Urban Landscape (CAUL) hub (funded by the Australian Government's Department of the Environment) set out to undertake an intercomparison of air quality models over New South Wales that would test existing capabilities, identify any problems and provide the necessary validation of models for the region. This project was designed to establish robust air quality modelling capabilities, by building on the substantial efforts of recent years by the Commonwealth Scientific and Industrial Research Organisation (CSIRO) and the New South Wales Department of Planning, Industry and Environment (DPIE), to improve the modelling of photochemical $O_3$ and secondary particle formation for air quality applications in the Sydney basin and surrounding areas through the development of the Conformal Cubic Atmospheric Model-Chemical Transport Model (CCAM-CTM) [5,19,20]. The modelling intercomparison tests the capabilities of six air quality modelling systems, including the DPIE's operational version of CCAM-CTM, against a number of other state-of-the-science air quality models including different versions of the widely used Weather Research and Forecasting with Chemistry (WRF-Chem) and Community Multi-Scale Air Quality (CMAQ) models. The skill of each model is assessed by comparing their simulation of the atmosphere against observations made from the DPIE's network of air quality monitoring stations in the Sydney basin during periods coinciding with three previous measurement campaigns:

1.  Sydney Particle Study stage 1 (SPS1) which took place in Westmead (33.80° S, 151.0° E), Sydney for ~4 weeks in summer from 5 February to 7 March 2011 [21,22];
2.  Sydney Particle Study stage 2 (SPS2) which took place in Westmead (33.80° S, 151.0° E), Sydney for ~4 weeks in autumn from 16 April to 14 May 2012 [22,23];
3.  Measurements of Urban Marine and Biogenic Air (MUMBA) which took place at the University of Wollongong's campus east (34.40° S, 150.9° E), Wollongong for 8 weeks in summer from 21 December 2012 to 15 February 2013 [24–28].

The performance of the models in representing meteorological conditions during the campaigns is presented in a separate paper [29], which showed:

1. The models overestimated wind speeds, especially at night time;
2. Temperatures were well simulated, with the largest biases also seen overnight;
3. The lower atmosphere was drier in the models than actually observed;
4. Meso-scale meteorological features, such as sea breezes were reproduced to some extent in the simulations [29].

Overall, the models generally performed within the recommended benchmark values for meteorology, except at high (and low) percentiles, when the biases tended to be larger [29].

The model simulations used for the intercomparison exercise were subsequently used in a number of additional studies, including benchmarking the performance of the DPIE's operational model [30] and using it to identify the major sources of $O_3$ [31] and $PM_{2.5}$ [32] in the greater Sydney region. Other studies explored the role of extreme temperature days in driving $O_3$ pollution events [33] and the relative performance of the WRF-Chem model with and without coupling to the Regional Ocean Model System [34,35].

In this paper, the performance of the models in representing ambient values of $O_3$ and $PM_{2.5}$ is evaluated. We first look at how the models reproduce the observed diurnal cycle in $O_3$ across all selected DPIE air quality monitoring sites. We then investigate the skill of the models at capturing the dominant $O_3$ formation regime (either limited by the availability of atmospheric volatile organic compounds (VOC limited) or by the availability of atmospheric nitrogen oxides ($NO_x$ limited)) at each air quality monitoring site. We also investigate the ability of the models at reproducing the timing and location of maximum daily $O_3$ values above 60 ppb. In addition, we assess the performance of the models at simulating 24 h average $PM_{2.5}$ concentrations at the DPIE air quality monitoring sites that measured $PM_{2.5}$ during the campaign periods. We also evaluate how well the models reproduce the chemical composition of the inorganic $PM_{2.5}$ fraction measured at the campaign sites.

## 2. Methods

### 2.1. Air Quality Modelling Systems

To simplify the presentation, each of the six modelling systems and their output will be referred to by a short acronym. This intercomparison examined three simulations based on the WRF-Chem model (W-UM2, W-NC1 and W-NC2), one simulation based on WRF-CMAQ (W-UM1) and two simulations based on CCAM-CTM (C-CTM and O-CTM). WRF-Chem is an online coupled regional-scale model [36] driven by the Advanced Research Weather Research and Forecasting (WRF) model [37]. WRF-Chem offers many options for physics, chemistry, and aerosols. All three WRF-Chem simulations are based on v3.7.1 of the model. However, W-NC1 and W-NC2 incorporate the developments described in Wang et al. (2015) [38] and simulate additional aerosol direct, semi-direct, and indirect effects that are not simulated in the other models. W-NC1 and W-NC2 use the same physics, chemistry, and aerosol options, but W-NC2 is coupled with the Regional Ocean Modelling System (ROMS) (WRF-Chem/ROMS) [39] and explicitly simulates air-sea interactions and sea-surface temperatures that are not simulated in W-NC1 or other models in this comparison [34,35].

The CMAQ model is an open-source chemistry-transport model developed and maintained by the US EPA [40]; W-UM1 used v5.0.2 of the model in an offline mode, driven by gridded meteorological fields from WRF v3.6.1.

The CCAM-CTM is a 3D Eulerian model developed for Australian regional air quality studies [19,41]. O-CTM is the operational version of the model run by the DPIE (previously NSW OEH) in New South Wales, whereas C-CTM is run by CSIRO. Both C-CTM and O-CTM derive their meteorology from the Cubic Conformal Atmospheric Model (CCAM; [42]). All information pertaining to the configuration of the meteorological models can be found in the companion paper "Evaluation of regional air quality models over Sydney, Australia: Part 1 Meteorological model comparison" by Monk et al., 2019 [29]. Further details pertaining to the configuration of the chemical transport modelling of each model run are presented in Table 1 and briefly described below.

**Table 1.** Summary of the details of the six air quality modelling systems used in the intercomparison.

| | Parameter | W-UM1 | W-UM2 | O-CTM | C-CTM | W-NC1 | W-NC2 |
|---|---|---|---|---|---|---|---|
| | Research group | Univ. melbourne | Univ. melbourne | DPIE | CSIRO | NCSU | NCSU |
| **Model specifications** | Met. model | WRF | WRF | CCAM | CCAM | WRF | WRF |
| | Chem. model | CMAQ | WRF-Chem | CSIRO-CTM | CSIRO-CTM | WRF-Chem | WRF-Chem-ROMS |
| | Met. model version | 3.6.1 | 3.7.1 | r−4271:4285M | r−2796 | 3.7.1 | 3.7.1 |
| | Chem. model version | 5.0.2 | 3.7.1 | r−1057 | r−1035 | 3.7.1 | 3.7.1 |
| **Domain** | Nests | 4 | 4 | 4 | 4 | 4 | 4 |
| | Horizontal res. (each nest) (km) | 81, 27,9,3 | 81, 27,9,3 | 80, 27,9,3 | 80, 27,9,3 | 81, 27,9,3 | 81, 27,9,3 |
| | Nx | 67, 60, 84, 90 | 80, 73, 97, 103 | 75, 60, 60, 60 | 75, 60, 60, 60 | 79, 72, 96, 102 | 79, 72, 96, 102 |
| | Ny | 57, 78, 84, 90 | 70, 91, 97, 103 | 65, 60, 60, 60 | 65, 60, 60, 60 | 69, 90, 96, 102 | 69, 90, 96, 102 |
| | Vertical layers | 29 | 33 | 16 | 16 | 32 | 32 |
| | Height of first layer (m) | 33.5 | 56 | ~20 | ~20 | ~35 | ~35 |
| **Chemical parametisations** | Gas-phase mechanism | CB05 with active chlorine chemistry, updated toluene mechanism | RACM with KPP (chem_opt = 105) | CB05, with updated toluene mechanism, precursors for VBS | CB05 with, updated toluene mechanism, precursors for VBS | CB05 with active chlorine chemistry | CB05 with active chlorine chemistry |
| | Aqueous-phase chemistry | AQChem | No aqueous phase chemistry | For sulfur | No aqueous phase chemistry | AQChem | AQChem |
| | Photolysis scheme | JPROC | fTUV | 2D photolysis based on Hough (1988) [43] | 2D photolysis based on Hough (1988) [43] | fTUV | fTUV |
| | Aerosol modules | Aero6 | MADE/SORGAM | 2-bin scheme | 2-bin scheme | MADE/VBS | MADE/VBS |
| | Inorganic aerosol thermodynamic module | ISORROPIA-II | MADE | ISORROPIA-II | ISORROPIA-II | ISORROPIA-II | ISORROPIA-II |
| | SOA module | Aero6 | SORGAM | VBS | VBS | VBS | VBS |
| | Dry deposition | Wesely (1989) scheme | Wesely (1989) [44] scheme | Wesely (1989) [44] scheme | Wesely (1989) [44] scheme (a) | Wesely (1989) [44] scheme (b) | Wesely (1989) [44] scheme (b) |
| | Wet deposition | Henry's law (gas phase), scavenging rate (aerosol in cloud water) | No wet deposition | Henry's law (gas phase), scavenging rate (aerosol in cloud water) | Henry's law (gas phase), scavenging rate (aerosol in cloud water) | (c) | (c) |

**Table 1.** *Cont.*

| | Parameter | W-UM1 | W-UM2 | O-CTM | C-CTM | W-NC1 | W-NC2 |
|---|---|---|---|---|---|---|---|
| | Research group | Univ. melbourne | Univ. melbourne | DPIE | CSIRO | NCSU | NCSU |
| **Emissions** | **Anthropogenic** | 2008 NSW GMR Air Emissions Inventory (d) | 2008 NSW GMR Air Emissions Inventory (d) | 2008 NSW GMR Air Emissions Inventory | 2008 NSW GMR Air Emissions Inventory | 2008 NSW GMR Air Emissions Inventory (d) | 2008 NSW GMR Air Emissions Inventory (d) |
| | **Biogenic** | MEGAN | MEGAN | ABCGEM | ABCGEM | MEGAN | MEGAN |
| | **Sea salt** | In-line | MADE/SORGAM | Clarke et al. (2003) [45] and Gong et al. (2003) [46] scheme | Clarke et al. (2003) [45] and Gong et al. (2003) [46] scheme | Gong et al. (1997) [47] scheme | Gong et al. (1997) [47] scheme |
| | **Dust** | In-line (wind blown) | In-line | Lu and Shao (1999) [48] scheme | Lu and Shao (1999) [48] scheme | AER/AFWA | AER/AFWA |
| | **Fire** | | | GFAS | | | |
| **Initial and boundary conditions** | **Chem. ICs/BCs** | MOZART | MOZART | Cape Grim observations and ACCESS_UKCA run | Cape Grim observations and ACCESS_UKCA run | CESM/CAM5 (1.2.2) (e) | CESM/CAM5 (1.2.2) (e) |

(a) Also refers to EPA (1999) [49]. (b) For all species except for sulfate; sulfate dry deposition based on Erisman et al. (1994) [50]; aerosol settling velocity and deposition based on Slinn and Slinn (1980) [51] and Pleim et al. (1984) [52]. (c) In-cloud wet removal of dissolved trace gases and the cloud-borne aerosol particles collected by rain, graupel, and snow (Grell et al., 2005) [36]. Below-cloud wet removal of aerosol particles by impaction scavenging via convective Brownian diffusion and gravitational or inertial capture, and irreversible uptake of $H_2SO_4$, $HNO_3$, HCl, $NH_3$, and simultaneous reactive uptake of $SO_2$, $H_2O_2$ (Easter, 2004) [53]. (d) EDGAR-HTAP (Janssens-Maenhout et al., 2012) [54] emissions used for domains not covered by Emissions Inventory. Volatile organic compound (VOC) speciation from the Intergovernmental Panel on Climate Change (IPCC) (2001) [55]. (e) with boundary conditions (BCONs) of $O_3$, $NO_2$, CO, and HCHO constrained based on satellite observations, and those for PM species were constrained based on MODIS AOD.

## 2.2. Air Quality Model Configuration

All models were run over four nested domains at horizontal grid resolutions of 81 (or 80 km for C-CTM and O-CTM), 27, 9 and 3 km. The outer-most domain (with the coarsest resolution) covers the whole of Australia whereas the inner-most domain covers the greater Sydney area. Figure 1 in [29] shows a map of the modelling domains. Although the models have consistent horizontal grids, they differ in their vertical resolution (16–35 layers) and in the height of their first model layer (~20–~56 m).

All models except W-UM2 used gas-phase chemistry mechanisms based on variations of CB05 [56]. W-NC1 and W-NC2 used a version with additional chlorine chemistry [57], whereas W-UM1, O-CTM and C-CTM used variants that included updated toluene chemistry [58,59]. W-UM2 used the Regional Atmospheric Chemistry Mechanism (RACM) of Stockwell et al. [60] with the Kinetic Preprocessor (KPP). This option does not permit the inclusion of the full WRF-Chem aqueous-phase chemistry, including aerosol–cloud interactions and wet scavenging [61].

All other models except C-CTM included aqueous chemistry: O-CTM included aqueous chemistry for sulfur [19] and W-NC1, W-NC2 and W-UM1 used the AQChem aqueous chemistry scheme from Sarwar et al. (2011) [62] implemented by Kazil et al. (2014) [63].

All three WRF-Chem simulations used the Fast Troposphere Ultraviolet-Visible (FTUV) photolysis model [64], whereas W-UM1 used clear-sky photolysis rates calculated offline using JPROC [65] and stored in look-up tables, and O-CTM and C-CTM used a 2D photolysis schemes based on Hough [43]. W-UM1, W-UM2, W-NC1 and W-NC2 used modal representations of particle size distribution, whereas C-CTM and O-CTM used a 2-bin ($PM_{2.5}$ and $PM_{10}$) sectional representation.

All models used a volatility basis set (VBS; [66–68] approach for SOA, except W-UM2 which used the Secondary Organic Aerosol Module (SORGAM; [69]) and W-UM1 which used the CMAQ Aero6 module [70,71]. All models incorporate version II of the ISORROPIA thermodynamic equilibrium module [72] for the treatment of inorganic aerosol, except W-UM2, which used the MADE scheme [73].

All models run in the experiment used the scheme described by Wesely [44] to handle dry deposition. W-NC1 and W-NC2 used the Wesely scheme for all species except sulfate. Sulfate dry deposition for these models were based on Erisman et al. [50], and aerosol settling velocity and deposition were based on Slinn and Slinn [51] and Pleim et al. [52]. The O-CTM/C-CTM resistive dry deposition scheme also refer to EPA (1999) [49].

W-UM2 did not include a wet deposition scheme. C-CTM and O-CTM used Henry's law for gas phase deposition and the scavenging rate for aerosol in cloud water. W-NC1, W-NC2 and W-UM1 took in-cloud wet removal of dissolved trace gases and the cloud-borne aerosol particles collected by hydrometeors [36,74]. Below-cloud wet removal of aerosol particles by impaction, scavenging via convective Brownian diffusion and gravitational or inertial capture, irreversible uptake of $H_2SO_4$, $HNO_3$, HCl, $NH_3$, and simultaneous reactive uptake of $SO_2$, $H_2O_2$ were also included [53].

## 2.3. Emissions

All models were coupled to the 2008 anthropogenic emissions inventory from the NSW EPA [75]. The inventory covers the NSW Greater Metropolitan Region (GMR), a region covering over 57,000 km$^2$ that includes Sydney, Newcastle and Wollongong. The inventory includes anthropogenic emissions of over 850 substances (including common pollutants such as CO, $NO_x$, $PM_{10}$, $PM_{2.5}$, $SO_2$, VOCs and greenhouse gases) from domestic, commercial and industrial sources, as well as on- and off- road sources. Emissions from licensed point sources are assigned to map coordinates whereas domestic, fugitive commercial and industrial, off- and on- road emissions are assigned to 1 km × 1 km grid cells. The emissions are then calculated for each month, day of week and hour of day, so that two sets of diurnal cycles are available for each month (weekday and weekend) for each 1 km × 1 km grid cell. Figure 1 shows the emission maps of anthropogenic $NO_x$ and $PM_{2.5}$, re-gridded to the 3 km model resolution, at 10:00 on a weekday in April. Although all models used the NSW EPA inventory, there were some slight differences in its implementation. Firstly, the inventory data had to be interpreted and made into model-ready files. This process was done separately for each modelling

system type (WRF-Chem, CMAQ, CTM). Also, C-CTM and O-CTM use Heating Degree Days (HDD) to normalise and scale domestic wood burning emissions, which results in a more realistic temporal release of woodburning emissions [6]. Finally, W-NC1, W-NC2, W-UM1 and W-UM2 used EDGAR emissions [54] in the parts of the domains not covered by the NSW EPA inventory.

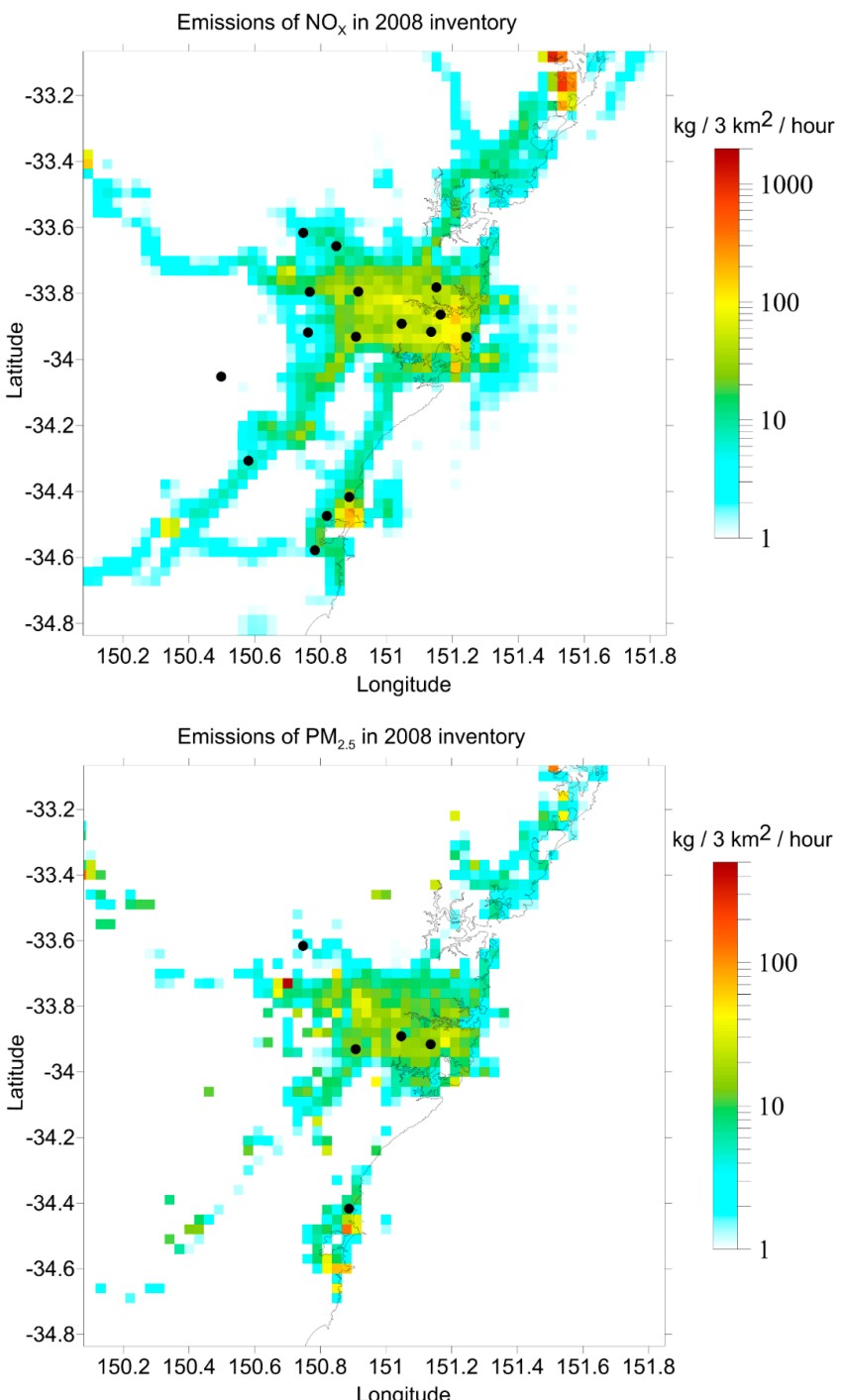

**Figure 1.** Example of $NO_x$ emissions (upper panel) and $PM_{2.5}$ emissions (lower panel) from the 2008 New South Wales (NSW) emissions inventory. Emissions are re-gridded to the 3 km model resolution and are for 10:00 on a weekday in April. Locations of the Department of Planning, Industry and Environment (DPIE) air quality monitoring stations measuring $O_3$ are shown as black dots in the upper panel, and those measuring $PM_{2.5}$ are shown as black dots in the lower panel.

VOC speciation in the WRF-Chem models was based on speciation in the "Prep_chem WRFChem" emission utility, from the Intergovernmental Panel on Climate Change (IPCC) speciation for VOCs [55]. All models applied PM ($PM_{2.5/10}$ and OC/EC) speciation developed by CSIRO. Biogenic emissions were generated online using the Model of Emissions of Gases and Aerosols from Nature (MEGAN v2.1) model [76,77] (W-UM1, W-UM2, W-NC1, W-NC2) or the Australian Biogenic Canopy and Grass Emissions Model (ABCGEM) [78] (C-CTM, O-CTM). Emissions of sea-salt aerosol and wind-blown dust were calculated online within the models using the parameterisations listed in Table 1 [45,46]. C-CTM also included fire emissions from GFAS [79], speciated according to Akagi et al. [80].

### 2.4. Initial and Boundary Conditions

The meteorological initial and boundary conditions (ICONs and BCONs) for W-NC1 and W-NC2 are based on the National Center for Environmental Prediction Final Analysis (NCEP-FNL) [81]. The chemical ICONs and BCONs are based on the results from a global Earth system model, the NCSU's version of the Community Earth System Model (CESM_NCSU) v1.2.2 [82–85]. The BCONs of CO, $NO_2$, HCHO, $O_3$, and PM species are constrained based on satellite retrievals. A more detailed description can be found in Zhang et al., 2019 [34].

Gas phase BCONs for ozone, methane, carbon monoxide, oxides of nitrogen, and seven VOC species including formaldehyde and xylene were taken from Cape Grim measurements [86] in the C-CTM model, while those for the aerosol phase were taken from a global ACCESS-UKCA model run [87]. The meteorological ICONs and BCONs for O-CTM are ERA-Interim global atmospheric reanalysis. The chemical ICONs and BCONs used by O-CTM are from the ACCESS-UKCA model [87], while W-UM1 and W-UM2 used boundary conditions from MOZART [88].

### 2.5. Observations

Models provided hourly output of surface trace gases and particulates for three time periods corresponding to the intensive measurement campaigns (SPS1, SPS2 and MUMBA) described earlier. The observations made at the campaign sites are supplemented by those of the DPIE monitoring network. The DPIE operates a network of air quality stations throughout the state of New South Wales. These stations provide measurements of pollutants including $O_3$, $NO_x$, $PM_{2.5}$ and $PM_{10}$. Measurements are continuously uploaded to a publicly accessible web page [89]. For this model evaluation, data from sixteen stations located in the greater Sydney region were selected. During the campaign periods, all sixteen stations reported hourly averages for $O_3$, $NO_x$ and $PM_{10}$ (see upper panel of Figure 1). Five of the stations also reported $PM_{2.5}$ (see lower panel of Figure 1 for their location). Model performance was evaluated separately for each campaign period.

## 3. Results and Discussion of Model Evaluation for $O_3$

### 3.1. Domain Average Model Performance for Hourly $O_3$

The Australian government specifies an hourly standard of ≤ 100 ppb and a 4 hourly standard of ≤ 80 ppb for $O_3$ in the National Environment Pollution Measure for Ambient Air Quality (NEPM) [90]. Analyses presented in this section use the hourly data averaged across all sixteen measurement sites reporting $O_3$ unless otherwise stated. The 4 hourly analysis is given in the Appendix A. The upper panel of Figure 2 shows the composite diurnal cycles of observed and modelled $O_3$ (averaging all the data from 16 sites from each hour of the day across every day of the campaign). The lower panel of Figure 2 shows the Taylor diagrams for average performance across the 16 sites for $O_3$.

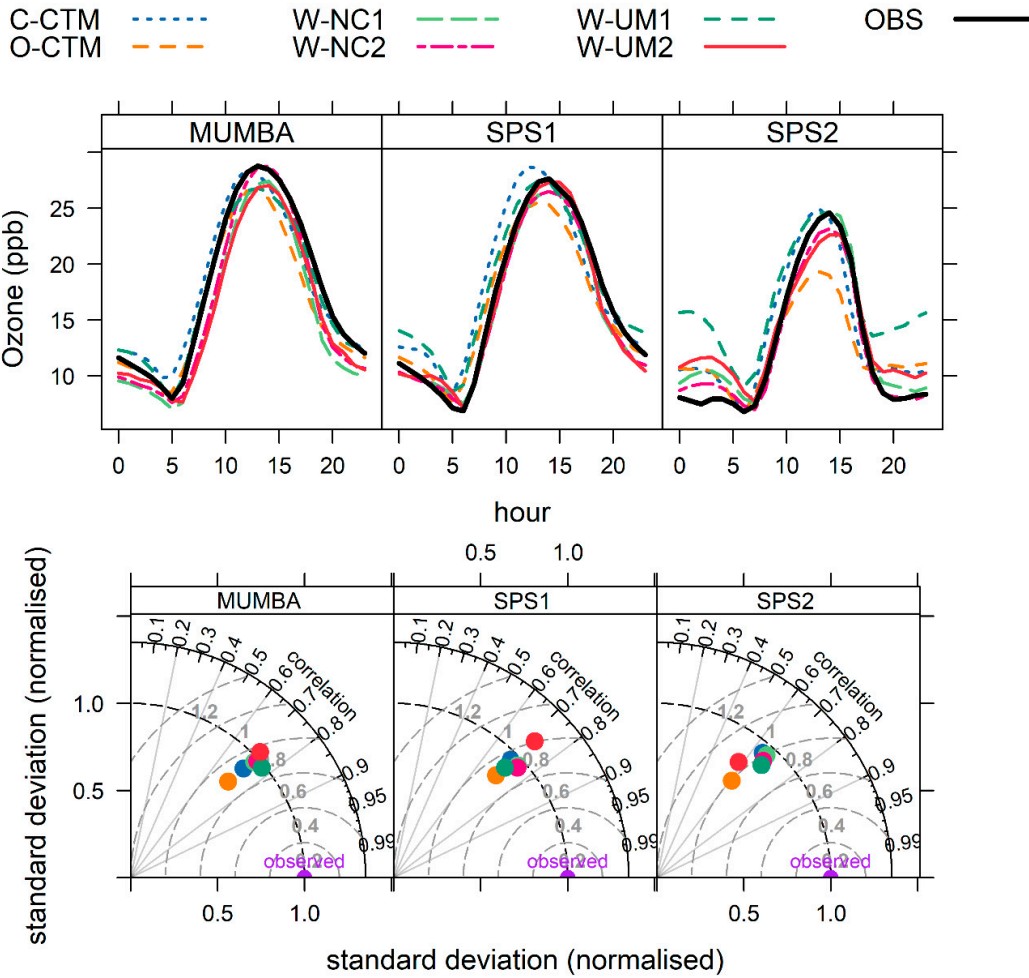

**Figure 2.** Composite diurnal cycles for observed and modelled $O_3$ during each campaign (upper panel) and Taylor diagrams for each campaign period (lower panel).

The models generally capture the observed $O_3$ diurnal cycle very well, especially in summer (MUMBA and SPS1). All models overestimate night-time $O_3$ in autumn (SPS2), and O-CTM also underestimates afternoon $O_3$ values during SPS2. For most models, the night-time overestimation of $O_3$ is likely caused by the underestimation of night-time $NO_x$ (see Figure A1), leading to insufficient titration. The Taylor diagrams in the lower panel of Figure 2 summarise model performance: The Pearson's correlation coefficient (r) between modelled and observed hourly variables is shown on the outside arc; the normalised standard deviation of the hourly observations is indicated as a dashed radial line (marked as 'observed' on the x axis); and the centred root mean squared error (RMSE) is indicated by concentric dashed grey lines emanating from the observed value. Overall the performance of the models is similar, in terms of correlation and RMSE, with a little more scatter in the standard deviation. All models underestimate the observed variability in $O_3$ during the SPS2 campaign.

Detailed performance statistics for each model for each campaign are given in Table 2. Absolute mean bias values of each model from the observations of $O_3$ are small (mean: ~1 ppb; max 4 ppb), but because mean $O_3$ levels are low, this translates to relatively high normalized mean bias (max 29%). A recent paper by Emery et al. [91] recommended goal and criteria values for the performance of photochemical models to predict $O_3$ amounts of < ± 5% (goal) and < ± 15% (criteria) for normalized mean bias (NMB) < 15% (goal) and < 25% (criteria) for normalized mean error (NME); and r > 0.75 (goal) and > 0.5 (criteria) for correlation.

**Table 2.** Summary statistics for $O_3$ are listed for each model and each campaign including mean and standard deviation (Sd); normalized mean bias (NMB); normalized mean error (NME) and correlation coefficient (r). Values for all the data are shown as well as for daytime data only (from 10:00 to 16:00 only).

| Campaign | Model | All Data | | | | | 10:00–16:00 Only | | | |
|---|---|---|---|---|---|---|---|---|---|---|
| | | Mean ± Sd (OBS) | Mean ± Sd (Model) | NMB % | NME % | r | Mean ± Sd (OBS) | Mean ± Sd (Model) | NMB | NME |
| MUMBA | C-CTM | 18 ± 12 | 18 ± 11 | 1.1 | 35 | 0.72 | 27 ± 13 | 27 ± 12 | −2.3 | 27 |
| | O-CTM | | 16 ± 9 | −7.7 | 34 | 0.71 | | 25 ± 10 | −8.6 | 27 |
| | W-NC1 | | 15 ± 12 | −13.6 | 37 | 0.73 | | 25 ±12 | −7.5 | 27 |
| | W-NC2 | | 16 ± 12 | −8.1 | 36 | 0.74 | | 26 ± 12 | −2.4 | 27 |
| | W-UM1 | | 17 ± 12 | −2.4 | 32 | 0.77 | | 26 ± 13 | −5.7 | 26 |
| | W-UM2 | | 16 ± 12 | −11.1 | 38 | 0.72 | | 25 ± 14 | −8.5 | 29 |
| SPS1 | C-CTM | 17 ± 11 | 18 ± 10 | 7.7 | 36 | 0.71 | 25 ± 10 | 27 ± 11 | 5.7 | 27 |
| | O-CTM | | 16 ± 9 | −2.4 | 35 | 0.71 | | 24 ± 9 | −5.4 | 24 |
| | W-NC1 | | 16 ± 10 | −4.6 | 35 | 0.74 | | 25 ± 10 | −3.4 | 25 |
| | W-NC2 | | 16 ± 10 | −4.6 | 34 | 0.75 | | 25 ± 9 | −3.5 | 24 |
| | W-UM1 | | 18 ± 11 | 7.1 | 36 | 0.71 | | 26 ±10 | 1.1 | 25 |
| | W-UM2 | | 16 ± 12 | −2.5 | 39 | 0.72 | | 25 ± 13 | −0.5 | 31 |
| SPS2 | C-CTM | 13 ± 10 | 14 ± 9 | 8.9 | 49 | 0.65 | 22 ± 7 | 22 ± 7 | −0.2 | 30 |
| | O-CTM | | 12 ± 7 | −3.2 | 50 | 0.64 | | 17 ± 6 | −20.6 | 32 |
| | W-NC1 | | 14 ± 9 | 6.7 | 46 | 0.67 | | 22 ± 7 | 2.1 | 22 |
| | W-NC2 | | 12 ±9 | −2.1 | 45 | 0.68 | | 21 ± 6 | −5.1 | 23 |
| | W-UM1 | | 17 ± 9 | 29 | 50 | 0.64 | | 23 ± 6 | 4.2 | 23 |
| | W-UM2 | | 14 ±8 | 7.8 | 52 | 0.63 | | 21 ± 6 | −5.9 | 24 |

For correlation, all models meet the criteria, and approach or reach the goal (> 0.75) for the summer campaigns. All models meet NMB criteria (< ± 15%) for all campaigns, except W-UM1 for SPS2. None of the models meet the criteria for NME, although the low mean $O_3$ amounts of < 20 ppb, make this a more difficult challenge than elsewhere in the world where $O_3$ amounts are typically significantly higher [14].

Emery et al. [91] recommended using a cut off of 40 ppb for the calculation of NMB and NME as a way to demarcate between nocturnal $O_3$ destruction (for which model performance is usually poor) and daytime $O_3$ production. In this study, this cut off is not applicable because it would exclude over 95% of observed values. Model performance is instead evaluated explicitly over $O_3$ production hours (10:00–16:00 local time). All models meet the NMB criteria over this subset of hours, except O-CTM during SPS2. NME values are generally improved over this subset of hours, especially for SPS2. W-NC1, W-NC2 and W-UM1 meet the NME criteria (< 25%) for SPS1 and SPS2.

Figure 3 shows quantile–quantile plots comparing modelled and observed hourly $O_3$ distributions for each campaign. In quantile–quantile plots the comparison is not a function of timing, but simply plots each quantile of model values against the corresponding quantile of observed values. Figure 3 shows that the models generally reproduce the observed $O_3$ distribution. However, there are deviations both at low quantiles (especially during SPS2) and high quantiles. For example, during MUMBA, C-CTM and O-CTM underestimated the higher hourly values, whereas W-UM1 and W-UM2 overestimated them.

When the model data are paired with the coincident observations, so that the timing in the models is important (Figure 4), all models overestimate low $O_3$ values and underestimate peak $O_3$ values. This indicates that the models do not capture the timing of low and high $O_3$ events. Similar results were noted in an evaluation of operational online-coupled regional air quality models over Europe and North America as part of the second phase of the Air Quality Model Evaluation International Initiative (AQMEII) [14].

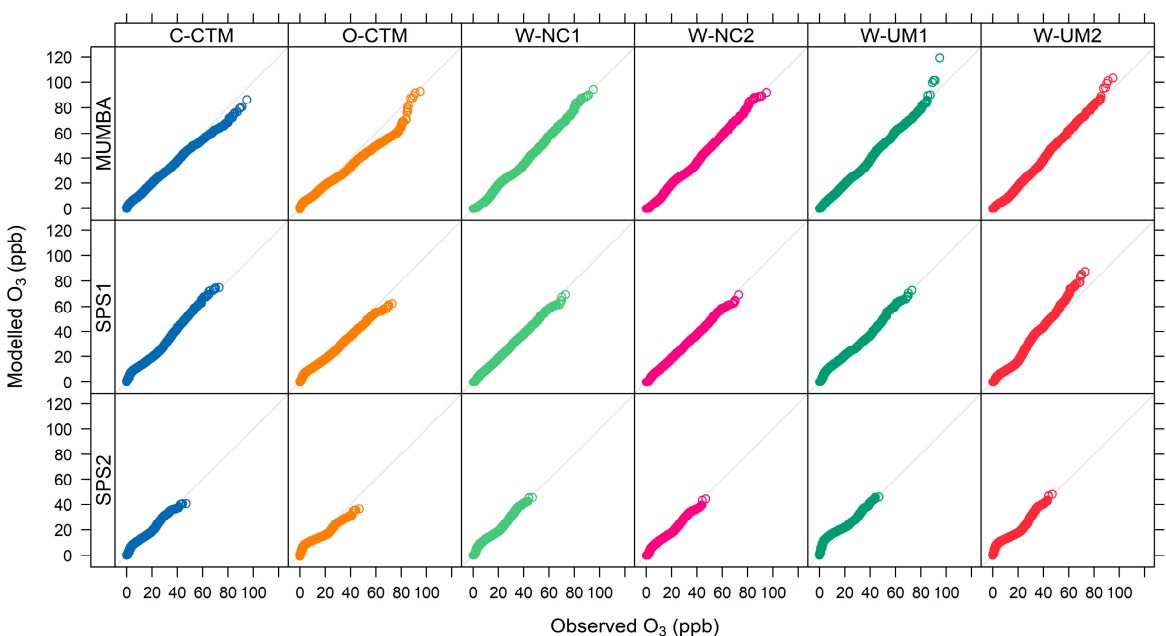

**Figure 3.** Quantile–quantile plot comparing modelled and observed hourly ozone distributions for each campaign. The x- and y- axes show ozone mixing ratios in ppb.

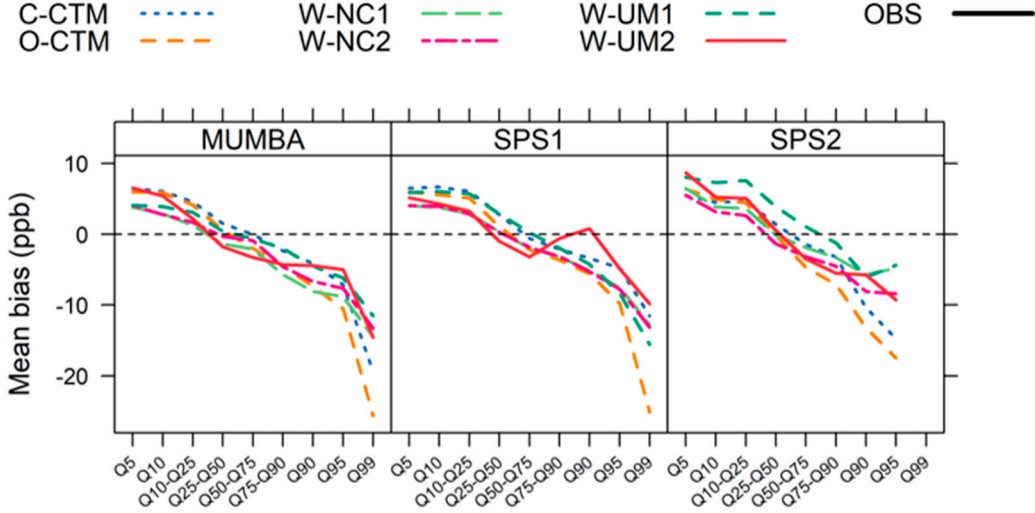

**Figure 4.** Mean bias for paired model/observed $O_3$ values, split into quantile bins (0–1, 1–5, 5–10, 10–25, 25–50, 50–75, 75–90, 90–95, 95–99 and 99–100 percentiles) for observed values.

*3.2. Domain Average Model Performance for 4 Hourly Average $O_3$*

Since the NEPM also includes a 4 hourly standard of less than 80 ppb for $O_3$ [90], the models were also evaluated for their performance for 4 h rolling means of $O_3$. See Table A2 for statistical results and Figure A1 for Taylor diagrams and mean bias for paired model/observed $O_3$ 4 hourly average values. The performance of the models for 4 h rolling means is slightly better than for hourly $O_3$. All models met the criteria for NMB. NME is smaller for all models and all campaigns. Correlation coefficients are improved for all models except W-NC1 and W-NC2. All models underestimate the observed variation in amplitude in 4 h rolling $O_3$ means, and do not capture the timing of high $O_3$ events (see Figure A1).

### 3.3. Site-Specific Model Performance for Hourly O₃

It is also useful to visualize model performance across the domain, to determine whether model performance is better (or worse) in some regions than others. The statistics listed in Table 2 reflect the average performance of the models across the 16 air quality monitoring sites. The maps in Figure 5 illustrate how model performance for correlation varies across the domain during each of the campaigns. Sites at which the goal is met (r > 0.75) are shown as triangles. Sites at which the goal is not met, but the correlation criteria (r > 0.5) is exceeded are shown as diamonds.

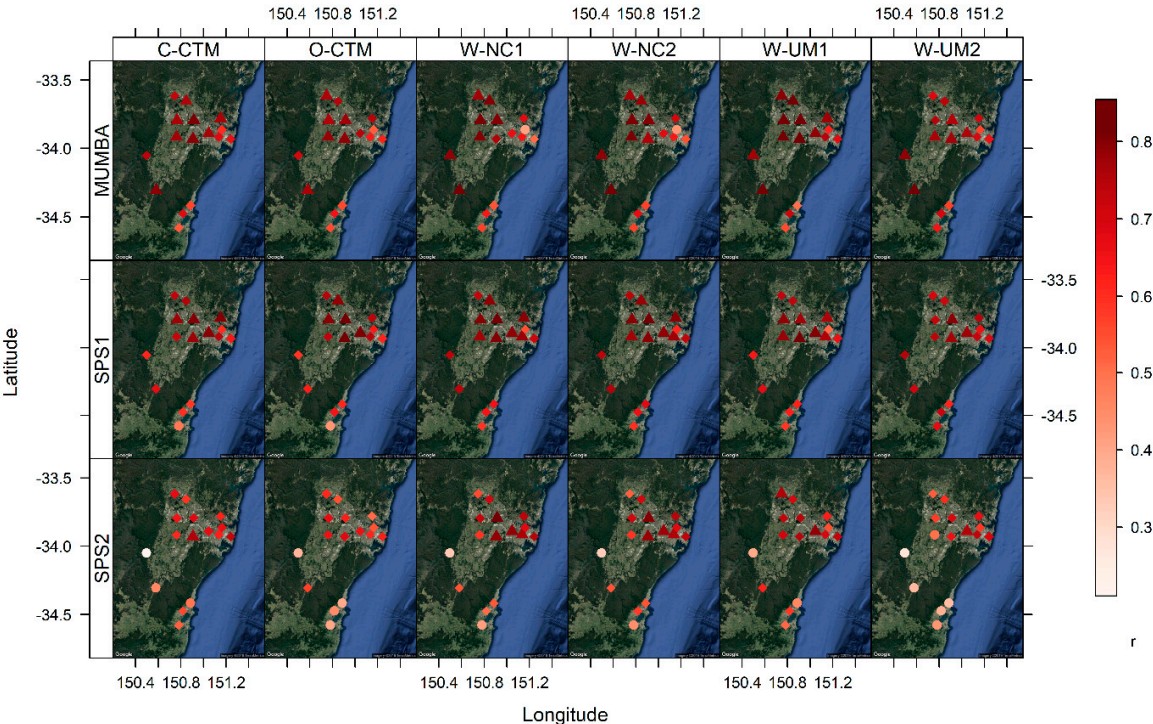

**Figure 5.** Map showing correlation performance between each model and the observed hourly O₃ values at the 16 DPIE air quality monitoring stations during the MUMBA, SPS1 and SPS2 campaigns. Sites at which the goal is met (r > 0.75) are shown as triangles. Sites at which the goal is not met, but the correlation criteria (r > 0.5) is exceeded are shown as diamonds.

The map indicates that the performance of the models is generally better in the northwest, with worse performance along the southern coast especially during SPS2.

### 3.4. Model Performance for Prediction of O₃ Pollution Events

An additional performance benchmark is whether models can accurately predict when high O₃ events (e.g., exceedances) occurred. This can be explored in terms of categorical statistics. In this analysis, we investigated the probability of detection (POD) of an O₃ event and the false alarm ratio (FAR) of each model for various O₃ thresholds. The POD is the ratio of the number of correctly predicted events over the number of observed events. The false alarm ratio is the number of incorrectly predicted events over the total number of predicted events.

Two thresholds were selected for the test: observed daily maximum O₃ > 60 ppb and observed daily maximum O₃ above 40 ppb (95th percentile of observed hourly O₃ values during the summer campaigns). We choose these thresholds to calculate the metrics instead of the regulatory standards because there was no exceedance of the hourly O₃ standard (100 ppb) during the modelled periods. Using daily maximum values instead of hourly values relaxes the test somewhat, as the exact timing of the high O₃ event does not need to be captured by the models.

The POD for daily maximum $O_3$ values > 60 ppb at any given site was generally poor (0%–67%); somewhat better (25%–80%) for prediction of a sub-regional event (e.g., Sydney East, Sydney North-West, etc.); and better still for prediction of an event somewhere within the domain (28%–93%). Relaxation of the geographical location of the predicted event from site specific, to regional and further to domain wide, greatly improved the number of false alarms, with the FAR decreasing each time the test was relaxed (false alarm ratio, domain wide: 10%–40%; region: 32%–73%; site: 40%–100%).

These results are not notably better than those found during a previous study that assessed the POD of $O_3$ events above 60 ppb in Melbourne and Sydney from December to March 2001–2002 and 2002–2003 using an earlier air quality model developed by CSIRO [92]. This earlier study found PODs of 23%–28% at individual sites; 21%–66% at the sub-regional scale and from 53% to 89%, at the domain scale [92]. It should be noted that there were significantly fewer peak ozone days in 2010–2011 (SPS1) and 2012–2013 (MUMBA) than there were during the periods of reference (2001–2002 and 2002–2003), and so the sample size in this current analysis is limited.

The results for prediction of $O_3$ events in this study are better when using a threshold of 40 ppb, with:

1.    Site-specific aggregated results: POD of 50%–83 % and FAR of 14%–46%
2.    Region-specific aggregated results: POD of 62%–86 % and FAR of 12%–35%
3.    Domain-wide results: POD of 67%–93 % and FAR of 4%–21%

These results are depicted graphically in Figure 6, with maps of the number of observed (left most column) and modelled events for daily maximum $O_3$ > 60 ppb (top panel) and > 40 ppb (bottom panel) for SPS1 and MUMBA. SPS2 is not shown due to a lack of events to display.

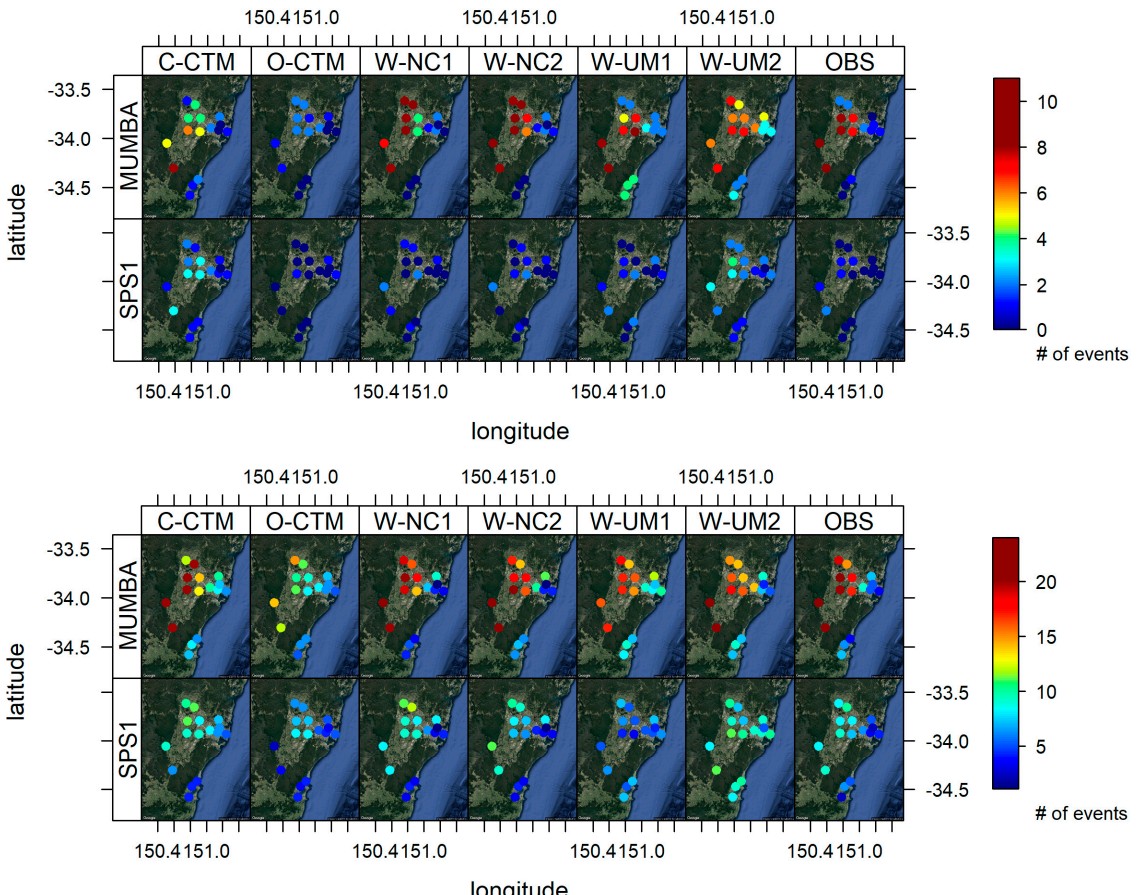

**Figure 6.** Maps of the number of observed (right-most column) and modelled events for daily maximum ozone > 60 ppb (**top panel**) and > 40 ppb (**bottom panel**).

### 3.5. Model Performance for Prediction of O$_3$ Production Regime

In this section, we evaluate whether the models reproduce the dominant observed O$_3$ production regime at each site, using O$_3$/NO$_X$ as the indicator [93]. This is important for guiding policies for reducing O$_3$ concentrations as it dictates whether a reduction in VOCs or NO$_X$ will result in an increase or decrease in O$_3$ in the region. The O$_3$/NO$_x$ ratio was calculated daily using values from 10:00–16:00 local time. At some stations during some of the campaigns, the NO$_x$ measurements were of too poor quality (with negative mixing ratios being reported) to reliably determine the O$_3$/NO$_x$ ratio. In these cases, the ratio was deemed unavailable at the site. A ratio < 15 was taken to indicate a VOC-limited O$_3$ production regime whereas values > 15 indicate a NO$_x$-limited regime [94], although the threshold values may vary as discussed in Zhang et al. [95]. The proportion of days with O$_3$/NO$_x$ < 15 (VOC-limited regime) was compiled for the observations and the models for each campaign and the results are shown in Figure 7.

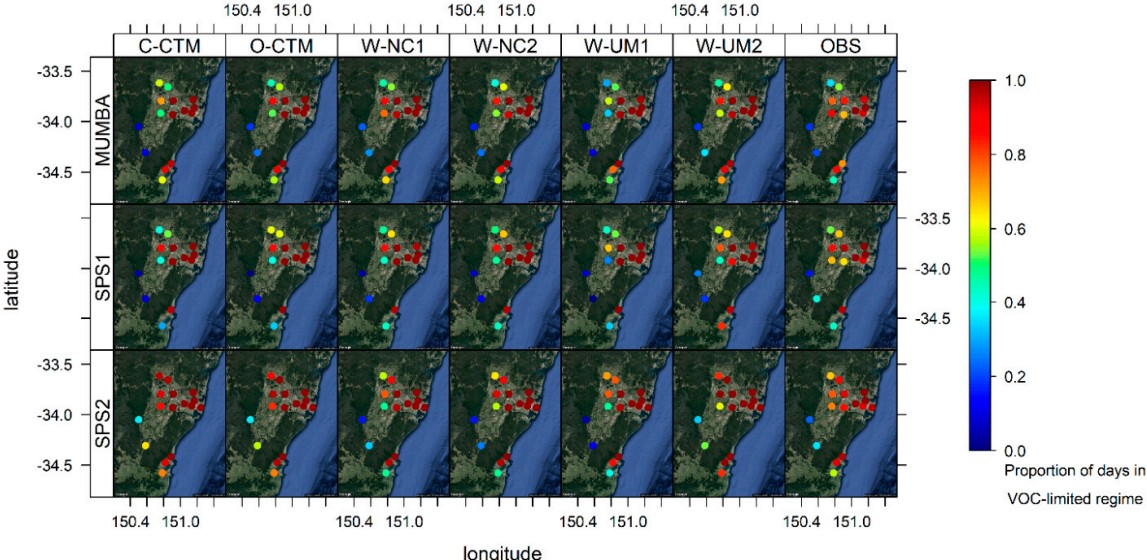

**Figure 7.** Maps showing proportion of days in VOC-limited O$_3$ production regime observed (right-most column) and in each of the models for each campaign.

During the SPS2 campaign, 14 out of the 16 sites experienced VOC-limited conditions (O$_3$/NO$_x$ < 15) on a majority of days. The models reproduce this pattern with a high level of accuracy, with all models predicting the right dominant regime at 13 or more of the sites. During SPS1, the O$_3$ regime indicator is available for 14 of the 16 sites. Of these, 11 experienced VOC-limited conditions on a majority of days. The models capture this pattern, with accurate predictions at 12 or more of the sites.

During MUMBA, the O$_3$ regime indicator is available for 15 of the 16 sites. Of these, 11 experienced VOC-limited conditions on a majority of days. Again, the models reproduce this pattern accurately, with all models predicting the right regime at 12 or more of the 15 sites.

This means that overall, modelled O$_3$ should respond in the appropriate way to increases/decreases in VOCs or NO$_x$.

## 4. Results and Discussion of Model Evaluation for PM$_{2.5}$

### 4.1. Domain Average Model Performance for Daily PM$_{2.5}$

The Australian government specifies a daily standard of ≤ 25 μgm$^{-3}$ and an annual standard of ≤ 8 μgm$^{-3}$ for PM$_{2.5}$ in the NEPM [90]. Model simulated PM$_{2.5}$ is evaluated against daily averaged observations at five sites for the summer (MUMBA, SPS1) periods and four sites for the autumn period

(SPS2). These sites used for evaluating PM$_{2.5}$ are all the available air quality monitoring sites that were making measurements of PM$_{2.5}$ in the modelled domain during the respective campaigns.

Composite time series of daily observed and modelled PM$_{2.5}$ and Taylor diagrams of model performance are shown for each campaign period in the two panels of Figure 8 and model performance statistics are given in Table 3. The plots show how model performance for PM$_{2.5}$ is more variable than for O$_3$. W-UM2 in particular is biased high during all campaigns, with much larger variability of PM$_{2.5}$ concentrations than seen in the observations. The low observed mean concentrations of PM$_{2.5}$ (e.g., 5.3 $\mu$gm$^{-3}$ during SPS2) mean that relatively small absolute differences become large normalized biases and errors, nevertheless model performance is generally much better for SPS1, during which the mean concentration of PM$_{2.5}$ was only marginally higher at 5.7 $\mu$gm$^{-3}$. One factor driving this worse performance for SPS2 is a much greater positive bias in W-NC1, W-NC2 and W-UM2 towards the latter end of SPS2 (see upper panel of Figure 8). Some of the bias may come from these models not applying scaling based on HDD to woodburning emissions; however, W-UM1 also does not use scaling but exhibit similar bias to C-CTM, which uses scaling. The use of EDGAR as a complementary inventory can also be dismissed as the cause of the gross overestimation by W-NC1, W-NC2 and W-UM2 since W-UM1 also uses EDGAR. Finally, there could have been errors in the preparation of the inventory files for May for the WRF-Chem modelling systems. Indeed, performance for PM$_{2.5}$ for all three models (W-NC1, W-NC2 and W-UM2) is much better in April than in May (e.g., NMB for W-NC1 is 292% in May but 68% in April).

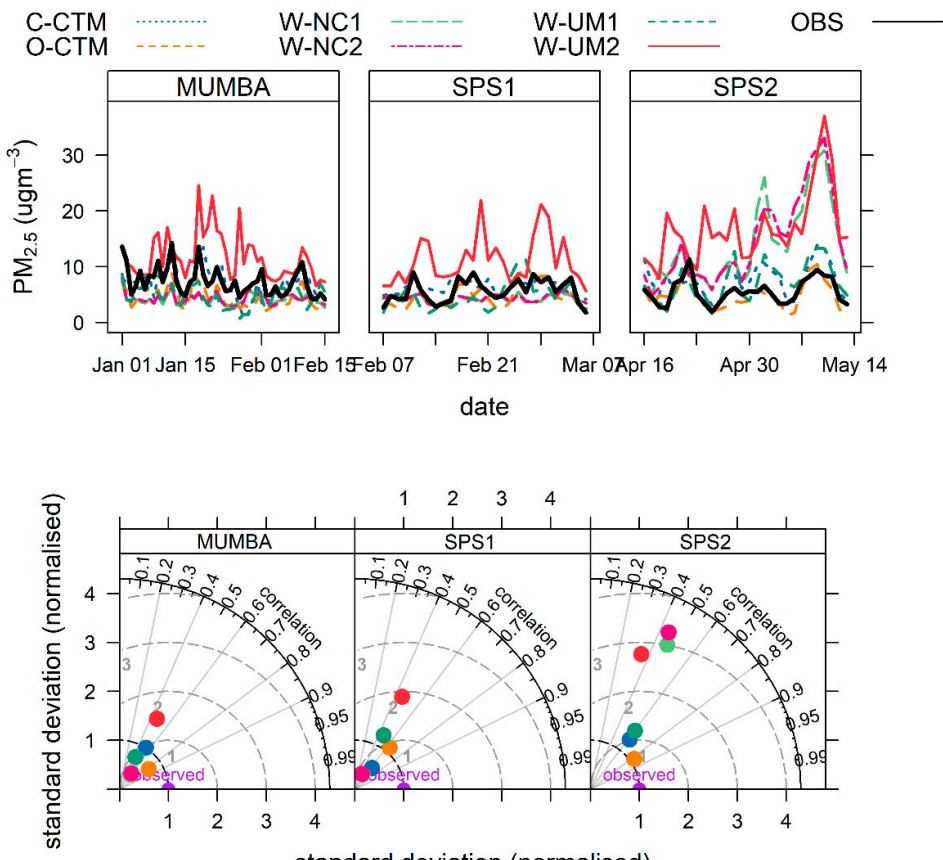

**Figure 8.** Composite time series of daily observed and modelled PM$_{2.5}$ during each campaign (upper panel) and Taylor diagrams for each campaign period (lower panel).

Emery et al. [91] recommended goal and criteria values for the performance of photochemical models to predict PM$_{2.5}$ amounts of < ± 10% (goal) and < ± 30% (criteria) for NMB; < 35% (goal) and < 50% (criteria) for NME; and r > 0.7 (goal) and > 0.4 (criteria) for correlation.

**Table 3.** Summary statistics for daily $PM_{2.5}$ concentrations in $\mu gm^{-3}$ are listed for each model and each campaign including mean and standard deviation (Sd); normalized mean bias (NMB); normalized mean error (NME) and correlation coefficient (r).

| Campaign | Model | Daily Means | | | | |
|---|---|---|---|---|---|---|
| | | Mean ± Sd (OBS) | Mean ± Sd (Model) | NMB % | NME % | r |
| MUMBA | C-CTM | 7.3 ± 2.7 | 6.9 ± 2.8 | −5.6 | 30 | 0.48 |
| | O-CTM | | 4.5 ± 2.0 | −39 | 40 | 0.72 |
| | W-NC1 | | 4.1 ± 1.3 | −44 | 45 | 0.47 |
| | W-NC2 | | 4.3 ± 1.3 | −42 | 43 | 0.48 |
| | W-UM1 | | 4.6 ± 2.1 | −37 | 43 | 0.37 |
| | W-UM2 | | 11.7 ± 4.6 | 59 | 63 | 0.45 |
| SPS1 | C-CTM | 5.7 ± 2.3 | 6.3 ± 1.3 | 11 | 29 | 0.54 |
| | O-CTM | | 4.8 ± 2.4 | −16 | 29 | 0.58 |
| | W-NC1 | | 4.3 ± 1.1 | −25 | 34 | 0.43 |
| | W-NC2 | | 4.3 ± 1.1 | −24 | 34 | 0.39 |
| | W-UM1 | | 4.6 ± 3.4 | −18 | 48 | 0.32 |
| | W-UM2 | | 11.1 ± 4.7 | 96 | 97 | 0.43 |
| SPS2 | C-CTM | 5.3 ± 2.8 | 7.5 ± 3.5 | 45 | 55 | 0.67 |
| | O-CTM | | 4.7 ± 3.1 | −8.0 | 30 | 0.79 |
| | W-NC1 | | 13.9 ± 13.4 | 170 | 177 | 0.58 |
| | W-NC2 | | 14.5 ± 13.9 | 180 | 186 | 0.57 |
| | W-UM1 | | 7.2 ± 4.4 | 38 | 57 | 0.61 |
| | W-UM2 | | 17.0 ± 9.6 | 227 | 230 | 0.51 |

Overall, model performance for $PM_{2.5}$ is worse than for $O_3$, but all models except W-UM2 meet the criteria (< 50%) for NME in summer (SPS1 and MUMBA), with some models meeting the goal (< 35%), especially during SPS1. All models meet the correlation criteria (> 0.4) during SPS2, and most do during the other campaigns. O-CTM meets the correlation goal (> 0.7) for SPS2 and MUMBA. The best performance for NMB is seen during SPS1, with all models except W-UM2 meeting the criteria (< ± 30%). All models fail to meet that criteria for the other two campaigns, except O-CTM during SPS2 and C-CTM during MUMBA; both models meet the goal (< ± 10%) in these instances.

Although not part of the regulatory framework for $PM_{2.5}$ in Australia, we also looked at the performance of the models for hourly $PM_{2.5}$. These results are presented briefly in the Appendix A: Figure A2 shows Taylor diagrams and composite diurnal cycles for observed and modelled hourly average $PM_{2.5}$ concentration during each campaign. Model performance for hourly average $PM_{2.5}$ is consistently worse than for daily average $PM_{2.5}$.

Figure 9 shows the quantile–quantile plots for domain averaged daily $PM_{2.5}$. This comparison removes the requirement for accurate timing, by plotting each quantile of model values against the corresponding quantile of observed values. C-CTM and O-CTM reproduce the observed $PM_{2.5}$ distribution quite well except for some low biases at the highest concentrations during MUMBA (O-CTM) and SPS1 (C-CTM). W-UM1 overestimates the higher $PM_{2.5}$ concentrations during SPS1 and SPS2 and underestimates them during MUMBA. W-NC1 and W-NC2 both underestimate the higher $PM_{2.5}$ concentrations in summer (MUMBA and SPS1) but overestimate $PM_{2.5}$ concentrations at all quantiles during SPS2. Finally, W-UM2 shows high biases across all three campaign periods.

*4.2. Site-Specific Model Performance for Daily $PM_{2.5}$*

The statistics listed in Table 3 reflect the average performance of the models across the five (or 4, for SPS2) air quality monitoring sites. The maps in Figure 10 illustrate how model performance for NMB varies across the domain. Sites at which the NMB criteria is exceeded (NMB < ± 30%) are shown as diamonds. Sites at which the goal is met (NMB < ± 10%) are shown as triangles. Figure 10 reveals that the poor performance during SPS2 is driven mostly by the very large biases seen at two of the sites

for W-NC1 and W-NC2, and at three of the sites for W-UM2. This highlights the problem of only having a small number of observational sites available to evaluate the models against in an intercomparison such as this. We note that the DPIE now measures $PM_{2.5}$ at all its monitoring sites, which will enable a much more detailed regional evaluation of model performance in future.

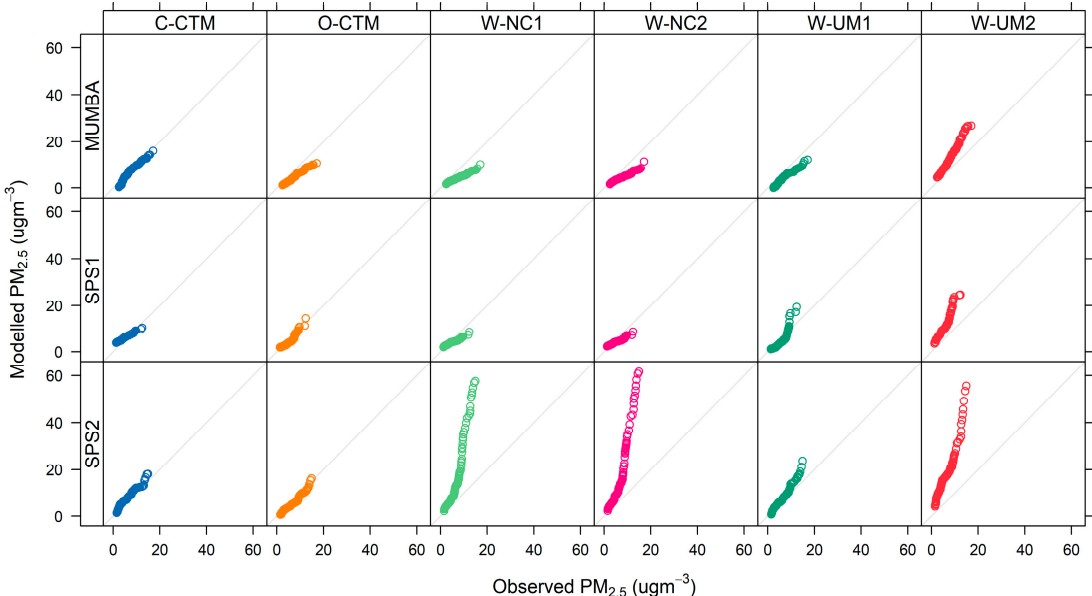

**Figure 9.** Quantile–quantile plot comparing modelled and observed distributions in daily $PM_{2.5}$ concentrations in $\mu gm^{-3}$ for each campaign.

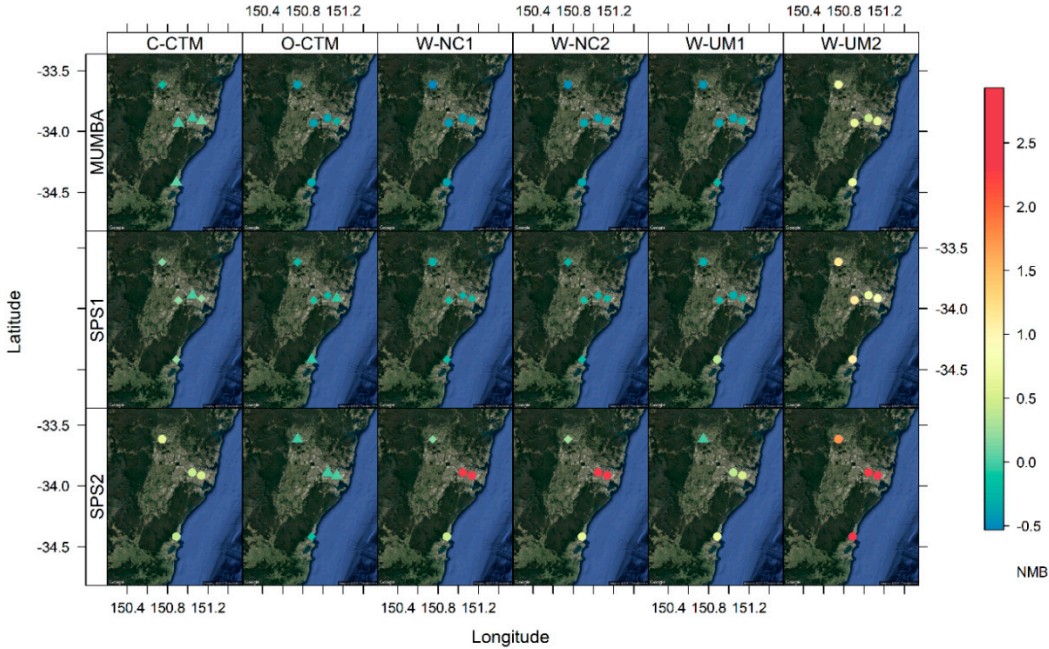

**Figure 10.** Model performance in terms of normalized mean bias (NMB) across the domain. Sites at which the NMB criteria is exceeded (NMB < ± 30%) are shown as diamonds. Sites at which the goal is met (NMB < ± 10%) are shown as triangles.

*4.3. Model Performance for $PM_{2.5}$ Inorganic Composition*

During each intensive measurement campaign (MUMBA, SPS1 and SPS2), measurements of the chemical composition of the inorganic fraction of $PM_{2.5}$ were made. $PM_{2.5}$ was collected onto filters

from 05:00 to 10:00 (from here on, denoted as morning or AM filters) and 11:00 to 19:00 (from here on, denoted as afternoon or PM filters) local time each day [21–23]. This allows for a limited (one site per campaign) evaluation of the performance of the models in predicting inorganic $PM_{2.5}$ composition, and to gain insight as to whether any particular fraction is contributing more to the model bias.

The model output was subsampled to match the timing of the observations. Figure 11 shows the median inorganic $PM_{2.5}$ concentration (in $\mu gm^{-3}$) from AM and PM filters across each campaign, coloured by its composite species: elemental carbon, sulfate ($SO_4^{2}$), nitrate ($NO_3^-$) and ammonium ($NH_4^+$).

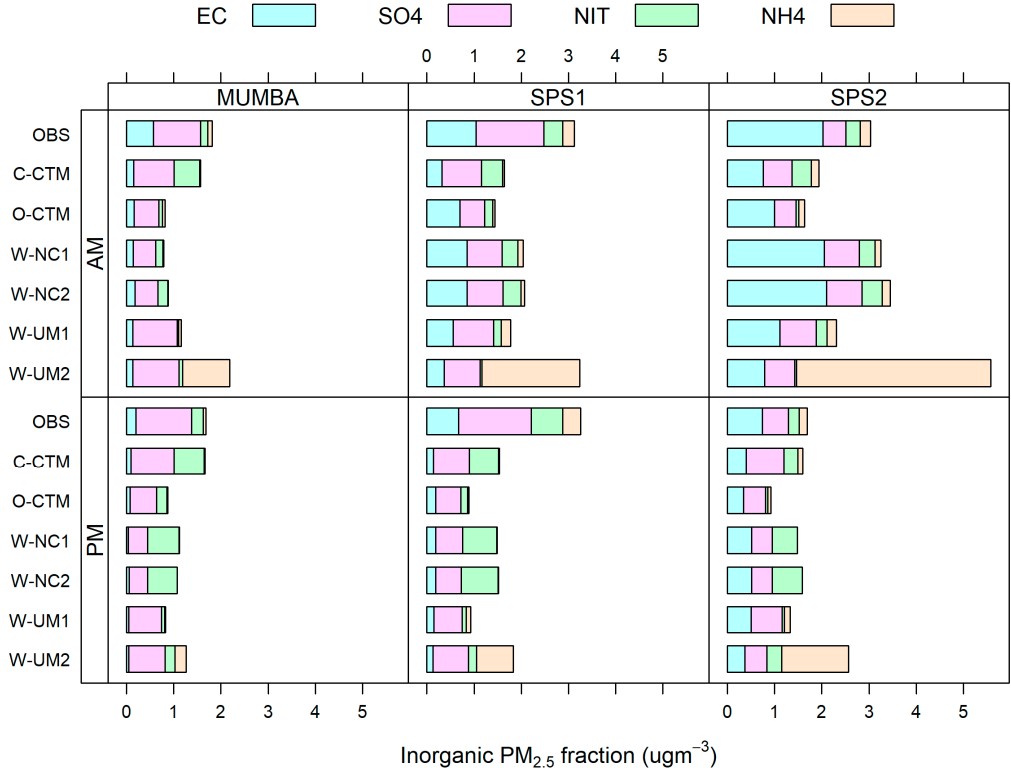

**Figure 11.** Median inorganic fraction of $PM_{2.5}$ in $\mu gm^{-3}$ from morning (AM) and afternoon (PM) filters, coloured by its composite species: elemental carbon (EC), sulfates ($SO_4$), nitrates (NIT) and ammonium ($NH_4$).

The figure shows that sulfate dominates the inorganic $PM_{2.5}$ during the summer campaigns (especially during MUMBA) and elemental carbon dominates in the autumn campaign (SPS2). Elemental carbon constitutes a significantly higher fraction of $PM_{2.5}$ on the AM filters, than the PM filters in all campaigns. Nitrate and ammonium typically make up only a small fraction of total inorganic $PM_{2.5}$, except for the PM filters in SPS1 where these two species together make up about one third of the total mass of $PM_{2.5}$. The models reproduce this median distribution fairly well; however, it is obvious from Figure 11 that W-UM2 overestimates $NH_4^+$. W-UM2 also predicts that very little ammonia ($NH_3$) remains in the gas phase (see Figure A4 for a box and whisker plot showing modelled and observed $NH_3$ values at the campaign sites). W-UM2 is the only model to use the MADE scheme instead of the ISORROPIA thermodynamic equilibrium module. Observed $NH_4^+$ levels are low on average ($< 0.35$ ug m$^{-3}$), and modelled values are within a factor of 2 of the observed values 11%–45% of the time. There is little nitrate observed ($< 0.7$ ug m$^{-3}$ on average) and modelled values are within a factor of 2 of observed values 15%–51% of the time. The models generally reproduce the observed difference in EC between the AM and the PM filters, but tend to underestimate EC in general. Modelled EC values are within a factor of 2 of the observed values 9 to 32% of the time in summer (MUMBA and

SPS1), and 48%–77% of the time in autumn (SPS2). Most models capture the sulfate contribution well, with 36%–83% of modelled values being within a factor of 2 of the observed values.

Inorganic $PM_{2.5}$ species contribute ~60% of the total mass of $PM_{2.5}$ during SPS1 and SPS2 and only 30% during MUMBA. This difference is probably linked to the location of the campaign sites: both SPS1 and SPS2 took place in Westmead in western Sydney where MUMBA took place in coastal Wollongong. The rest of the $PM_{2.5}$ mass likely comes from sea salt, dust and organic carbon (both primary and secondary). A more comprehensive evaluation would need to include these additional species. Most of the models underestimate total inorganic $PM_{2.5}$ loading in summer, irrespective of campaign site, which may contribute to the underestimation of total $PM_{2.5}$ mass by most models seen in Table 3 for MUMBA and SPS1. Figure 11 also indicates that the overestimation in total $PM_{2.5}$ seen in most models during SPS2 is not due to a gross overestimation of the inorganic fraction; however, the analysis presented in Figure 11 only covers daytime, whereas the worse overestimation of $PM_{2.5}$ occurs overnight (see Figure A3).

## 5. Summary and Conclusions

This paper presents the results of an intercomparison study to test the performance of six air quality modelling systems in predicting $O_3$ and $PM_{2.5}$ concentrations in Sydney and the surrounding metropolitan areas. Model performance for $O_3$ was evaluated against measurements at 16 air quality monitoring stations, whilst observations of $PM_{2.5}$ were only available from five stations (four during SPS2). Overall domain-wide hourly $O_3$ predictions by the models were accurate, and the observed $O_3$ production regime (based on the $O_3/NO_x$ indicator) reproduced at 80% or more of the air quality monitoring sites. The models generally capture the observed $O_3$ diurnal cycle very well, especially in summer. The models also generally met benchmark criteria for correlation (of greater than 0.5) and NMB (of less than 15%) as proposed by Emery et al. [91], despite overestimation of the lowest and underestimation of the highest observed hourly $O_3$ values. Model performance was better in the northwest, with poorer performance along the southern coast.

The probability of detection of $O_3$ events was better for a threshold of 40 ppb than for a threshold of 60 ppb. For both thresholds, the performance of the models improved (with the probability of detection increasing and the false alarm ratio decreasing) each time the geographical location criteria of the predicted event were relaxed from site specific, to regional, to domain wide. Further improvements in modelling systems are necessary to provide more accurate site-specific $O_3$ forecasts, including advances in the inventory of precursor emissions.

Domain-wide model performance for daily $PM_{2.5}$ was variable, with most models underestimating summer and overestimating autumn $PM_{2.5}$ concentrations. All models met the criteria for correlation (> 0.4) during the autumn campaign and most did during the summer campaigns. The benchmark criteria for NMB (< 30%) was met by only one model during MUMBA (C-CTM) and SPS2 (O-CTM), but by most models during SPS1. Analysis of the composition of the inorganic fraction of $PM_{2.5}$ showed that sulfate dominated in summer campaigns and elemental carbon dominated in the autumn campaign, with higher amounts of elemental carbon in the mornings. The models reproduced the dominant sulfate contribution, underestimated the morning elemental carbon and performed variably for nitrate and ammonium.

The relatively low pollution levels for $O_3$ and $PM_{2.5}$ in Sydney mean that a small absolute bias translates into a relatively large normalized bias, making the benchmark values set by Emery et al. [91] especially challenging. The small number of monitoring sites reporting $PM_{2.5}$ at the time of the campaigns is an additional challenge for the evaluation of the performance of the models for $PM_{2.5}$. Nevertheless, the modelling comparison exercise described in this paper has produced improvements in the implementation of these six models for New South Wales, benchmarked their performance against international standards and thereby increased confidence in their ability to simulate atmospheric composition within the greater Sydney region.

**Author Contributions:** Conceptualization, C.P.-W., Y.S., P.J.R. and M.E.C.; methodology, E.-A.G., K.M, J.D.S., K.M.E., S.R.U., Y.Z. and L.T.-C.C.; formal analysis, E.-A.G.; investigation, E.-A.G.; data curation, E.-A.G., K.M., J.D.S., K.M.E., S.R.U., Y.Z., L.T.-C.C., H.N.D. and T.T.; writing—original draft preparation, E-.A.G., C.P.-W. and J.S.; writing—review and editing, all authors; visualization, E-.A.G.; supervision, C.P.-W.; project administration, C.P.-W.; funding acquisition, C.P.-W., Y.S., P.J.R. and M.E.C. All authors have read and agreed to the published version of the manuscript.

**Funding:** This research was funded by Australia's National Environmental Science Program through the Clean Air and Urban Landscapes hub. YZ acknowledges the support by the University of Wollongong (UOW) Vice-Chancellors Visiting International Scholar Award (VISA), the University Global Partnership Network (UGPN), and the NC State Internationalization Seed Grant. Simulations using W-NC1 and W-NC2 were performed on Stampede and Stampede 2, provided as an Extreme Science and Engineering Discovery Environment (XSEDE) digital service by the Texas Advanced Computing Centre (TACC), and on Yellowstone (ark:/85065/d7wd3xhc) provided by NCAR's Computational and Information Systems Laboratory, sponsored by the National Science Foundation.

**Acknowledgments:** 

**Conflicts of Interest:** The authors declare no conflict of interest.

## Appendix A

The appendix contains:

1. Supplementary information about the meteorological setup of the models (see Table A1)
2. Analysis of the performance of the models for NOx, including composite diurnal cycles for observed and modelled hourly average NOx concentration in ppb and Taylor diagrams for each campaign period (see Figure A1).
3. Additional analysis of the performance of the models for $O_3$ on a 4 hourly basis rather than the 1 hourly basis presented in the main manuscript, including statistic (see Table A2) and Taylor diagrams and mean bias for paired model/observed (see Figure A2)
4. Additional analysis of the performance of the models for hourly average $PM_{2.5}$, including composite diurnal cycles for observed and modelled hourly average $PM_{2.5}$ concentration in $\mu gm^{-3}$ and Taylor diagrams from each campaign period (see Figure A3).
5. Analysis of the performance of the models for ammonia, in the form of a box and whisker plot (see Figure A4).

**Table A1.** Overview of the configuration of the meteorological models—reproduced from the companion paper "Evaluation of Regional Air Quality Models over Sydney and Australia: Part 1—Meteorological Model Comparison" [29].

| | Parameter | W-UM1 | W-UM2 | W-A11 | O-CTM | C-CTM | W-NC1 | W-NC2 |
|---|---|---|---|---|---|---|---|---|
| **Model Identifier** | Research group | Univ. melbourne | Univ. melbourne | ANSTO | NSW OEH | CSIRO | NCSU | NCSU |
| **Model specifications** | Met. model | WRF | WRF | WRF | CCAM | CCAM | WRF | WRF |
| | Chem. model | CMAQ | WRF-Chem | WRF-Chem with simplified Radon only | CSIRO-CTM | CSIRO-CTM | WRF-Chem | WRF-Chem-ROMS |
| | Met model version | 3.6.1 | 3.7.1 | 3.7.1 | r−3019 | r−2796 | 3.7.1 | 3.7.1 |
| **Domain** | Nx | 80,73,97,103 | 80,73,97,103 | 80, 73, 97, 103 | 75, 60, 60, 60 | 88, 88, 88, 88 | 79, 72, 96, 102 | 79, 72, 96, 102 |
| | Ny | 70,91,97,103 | 70,91.97.103 | 70, 91, 97, 103 | 65, 60, 60, 60 | 88, 88, 88, 88 | 69, 90, 96, 102 | 69, 90, 96, 102 |
| | Vertical layers | 33 | 33 | 50 | 35 | 35 | 32 | 32 |
| | Height of first layer (m) | 33.5 | 56 | 19 | 10 | 20 | 35 | 35 |
| **Initial and Boundary conditions** | Met input/BCs | ERA Interim | ERA Interim | ERA Interim | ERA Interim | ERA Interim | NCEP/FNL | NCEP/FNL |
| | Topography/Land use | Geoscience Australia DEM for inner domain, USGS elsewhere | Geoscience Australia DEM for inner domain. USGS elsewhere | Geoscience Australia DEM for inner domain, USGS elsewhere. MODIS land use | MODIS | MODIS | USGS | USGS |
| | SST | High-res SST analysis (RTG_SST) | High-res SST analysis (RTG_SST) | High-res SST analysis (RTG_SST) | SST from ERA Interim | SSTs from ERA Interim | High-res SST analysis (RTG_SST) | Simulated by ROMS |
| | Integration | 24 h simulations, each with 12 h spin-up number | Continuous with 2D spin up | Continuous with 10 d spin up | Continuous with 1 mth spin up | Continuous with 1 mth spin up | Continuous with 8 d spin up | Continuous with 8 d spin up |
| | Data assimilation | Grid-nudging outer domain above the PBL | Grid-nudging outer domain above the PBL | Spectral nudging in domain 1 above the PBL (scale-selective relaxation to analysis) | Scale-selective filter to nudge towards the ERA-Interim data | Scale-selective filter to nudge towards the ERA-Interim data | Gridded analysis nudging above the PBL | Gridded analysis nudging above the PBL |
| **Parameterisations** | Microphysics | Morrison | LIN | WSM6 | Prognostic condensate scheme | Prognostic condensate scheme | Morrison | Morrison |
| | LW radiation | RRTMG | RRTMG | RRTMG | GFDL | GFDL | RRTMG | RRTMG |
| | SW radiation | RRTMG | GSFC | RRTMG | GFDL | GFDL | RRTMG | RRTMG |
| | Land surface | NOAH | NOAH | NOAH | Kowalczyk scheme | Kowalczyk scheme | NOAH | NOAH |
| | PBL | MYJ | YSU | MYJ | Local Richardson number and non-local stability | Local Richardson number and non-local stability | YSU | YSU |
| | UCM | 3-category UCM | NOAH UCM | Single layer UCM | Town Energy budget approach | Town Energy budget approach | Single layer UCM | Single layer UCM |
| | Convection | G3 (domains 1–3, off for domain 4) | G3 | G3 | Mass-flux closure | Mass-flux closure | MSKF | MSKF |
| | Aerosol feedbacks | No | No | No | Prognostic aerosols with direct and indirect effects | Prognostic aerosols with direct and indirect effects | Yes | Yes |
| | Cloud feedbacks | No | No | No | Yes | Yes | Yes | Yes |

**Table A2.** Summary statistics for $O_3$ 4 hourly average values are listed for each model and each campaign including mean and standard deviation (Sd); normalized mean bias (NMB); normalized mean error (NME) and correlation coefficient (r).

| Campaign | Model | 4 Hourly Rolling Means | | | | |
|---|---|---|---|---|---|---|
| | | Mean ± Sd (OBS) | Mean ± Sd (Model) | NMB % | NME % | r |
| MUMBA | C-CTM | 18 ± 11 | 17 ± 10 | −6.2 | 29 | 0.79 |
| | O-CTM | | 17 ± 10 | −5.0 | 31 | 0.79 |
| | W-NC1 | | 16 ± 10 | −6.9 | 33 | 0.72 |
| | W-NC2 | | 17 ± 10 | −6.4 | 31 | 0.75 |
| | W-UM1 | | 16 ± 10 | −7.8 | 28 | 0.80 |
| | W-UM2 | | 16 ± 11 | −8.6 | 28 | 0.81 |
| SPS1 | C-CTM | 17 ± 10 | 17 ± 9 | 2.0 | 30 | 0.77 |
| | O-CTM | | 17 ± 9 | 2.6 | 34 | 0.71 |
| | W-NC1 | | 16 ± 9 | −0.8 | 34 | 0.71 |
| | W-NC2 | | 17 ± 9 | −0.2 | 33 | 0.73 |
| | W-UM1 | | 16 ± 9 | −1.1 | 28 | 0.80 |
| | W-UM2 | | 16 ± 9 | −1.1 | 28 | 0.80 |
| SPS2 | C-CTM | 13 ± 9 | 14 ± 8 | 13.1 | 43 | 0.69 |
| | O-CTM | | 14 ± 7 | 10.6 | 45 | 0.65 |
| | W-NC1 | | 13 ± 8 | 2.5 | 44 | 0.65 |
| | W-NC2 | | 13 ±7 | 2.7 | 44 | 0.65 |
| | W-UM1 | | 14 ± 7 | 7.5 | 39 | 0.72 |
| | W-UM2 | | 14 ±8 | 10.3 | 42 | 0.70 |

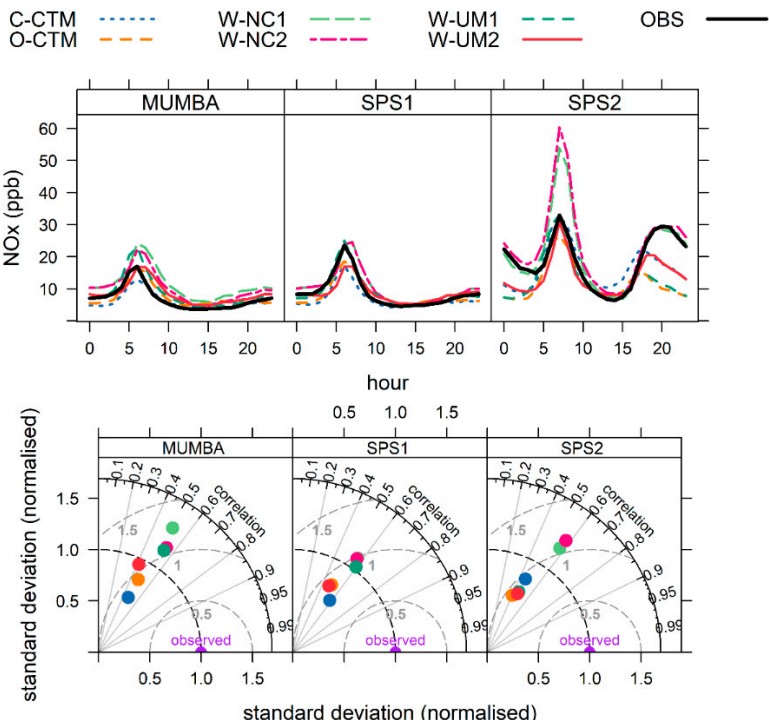

**Figure A1.** Composite diurnal cycles for observed and modelled hourly average $NO_x$ concentration in ppb during each campaign (upper panel) and Taylor diagrams for hourly $NO_x$ from each campaign period (lower panel).

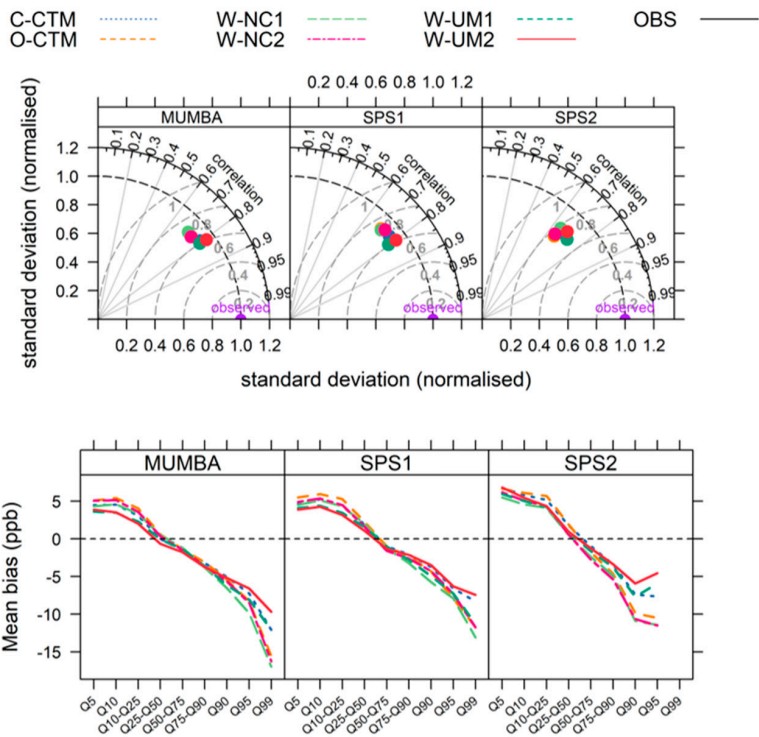

**Figure A2.** Taylor diagrams for $O_3$ 4 hourly average values for each campaign period (upper panel) and mean bias for paired model/observed $O_3$ 4 hourly average values, split into quantile bins (0–1, 1–5, 5–10, 10–25, 25–50, 50–75, 75–90, 90–95, 95–99 and 99–100 percentiles) for observed values (lower panel).

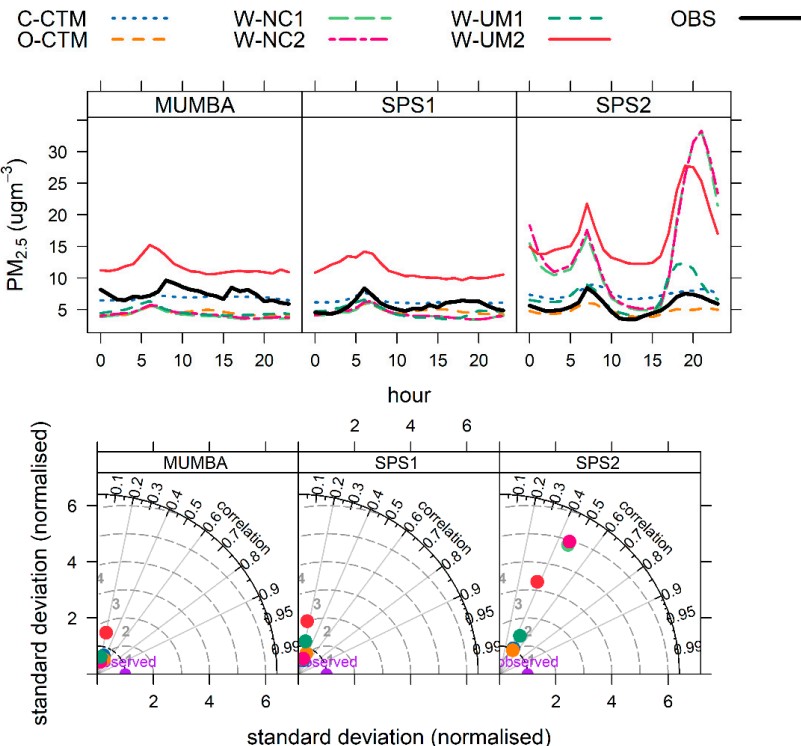

**Figure A3.** Composite diurnal cycles for observed and modelled hourly average $PM_{2.5}$ concentration in $\mu gm^{-3}$ during each campaign (upper panel) and Taylor diagrams for hourly $PM_{2.5}$ from each campaign period (lower panel).

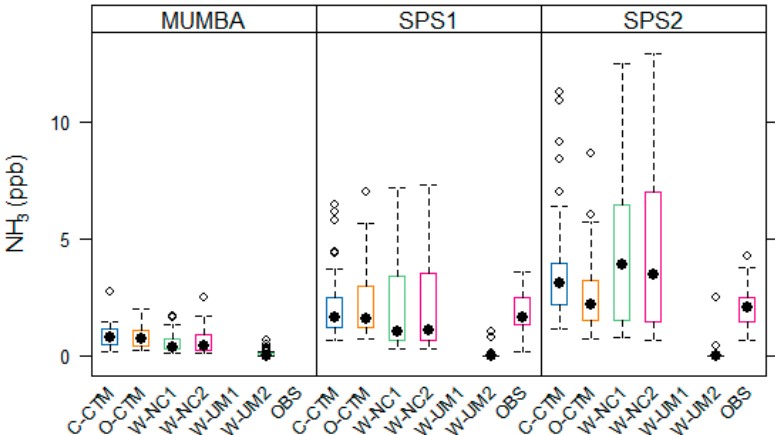

**Figure A4.** Box and whisker plots showing observed and modelled ammonia ($NH_3$) at each campaign site. The black dots are the average values, the box marks the first and third quartiles and the whiskers extend up to 1.5 length of the box. Outliers are open circles. No observations are available for MUMBA and no output is available from W-UM1.

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
