# Peer review of "Evaluation of Regional Air Quality Models over Sydney, Australia: Part 2, Comparison of PM2.5 and Ozone"

_atmosphere, doi:10.3390/atmos11030233_

Round 1

Reviewer 1 Report

Guerette et al. present a model comparison for O3 and PM 2.5 over three measurement campaign periods for Sydney, Australia. The six models represent a variety of meteorology, chemistry, source emissions and partitioning schemes. I recommend publication after flushing out a few minor points.

Major Comments:

There are a couple of places that appears to be lacking somewhat in supporting information:

Section 4.2: After comparing the model to the measurements of PM 2.5, they state “ This highlights the problem of only having a small number of observational sites available to evaluate the models against in an intercomparison such as this.” While the authors are correct in this, they don’t say anything further about it, or probe what the appropriate number of measurement sites would be for a comparison of this nature. Or even if there is someone else that has already done work quantifying this that could be cited would do.

Line 543: “probably attributable” is too loose and could use more explanation of why or how overlap or lack thereof of the campaign sites would influence differences in model performance.

Line 570: Here you state that these newer models showed no improvement over models from the early 2000’s.

Line 583: Here you state that only one model met the bench mark criteria during two of the campaigns, but not explanations are given for why that one model would perform better.

Minor Corrections:

Be consistent with abbreviations (when to use them and when not to), for instance in the Summary and Conclusions (lines 565 and 583) NMB and normalized mean bias are used.

Appendix Table A1: The second column contains several lines where a single letter is carried over to the next line (or a few letters) and make the information difficult to read. Consider ways to fix that formatting issue.

Author Response

We thank the reviewer for taking the time to read our manuscript and for their comments.

Please see the attachment for our responses.

Reviewer 2 Report

Title: Evaluation of regional air quality models over Sydney, Australia: Part 2, comparison of PM2.5 and ozone

Authors: Elise-Andrée Guérette *, Lisa Tzu-Chi Chan, Martin E. Cope, Hiep N. Duc, Kathryn M. Emmerson, Khalia Monk, Peter Rayner, Yvonne Scorgie, Jeremy D. Silver, Steven R. Utembe, Jack Simmons, Toan Trieu, Yang Zhang, Clare Paton-Walsh

This manuscript is a second part of a study evaluating air quality models performance. In the first part, the authors presented meteorological model comparison, and in the current part is the performance of the models in representing values of O3 and PM2.5 in ambient air in Sydney, Australia. In order to evaluate the models performance for O3 and PM2.5 representation, the authors selected five evaluation steps: they investigated how the models reproduced observed cycles of O3 at the monitoring sites, how was captured the dominant O3 formation regime, how models reproduced maximum daily O3 values, how was the performance of the models at simulating hourly and 24-hr average PM2.5 concentrations, and chemical composition of the inorganic fraction. The performance of six air quality models was assessed by comparing their simulations with measurements at monitoring stations in summer (SPS1, MUMBA) and fall (SPS2).  

I read this manuscript with interest. This is thoroughly carried, discussed, and well written study. The topic aligns with the scope of Atmosphere. Although evaluating air quality models is a popular topic and there are many similar published studies, this particular study seems to be unique in sense of evaluating 1-hr, 4-hr, and 24-h air quality models, not undertaken in the Sydney region and providing for standard/guidelines needs. This study is therefore important for feasibility of using such models for air quality forecasts and providing scientific data for discussions on policies regulating contributions to O3 and PM2.5 ambient levels in the Sydney region. A shortcoming in this study is the lack of intensive measurement campaign during winter and spring that observations could be used against model predicted values in better evaluation of the performance throughout the year as the model performance results suggest seasonal fluctuations. I recommend this manuscript for publishing with some minor corrections.

Minor corrections:

1.This is a long paper with many tables and figures. I suggest word counting to see if authors did not exceed the journal limits.

2.The authors tested six air quality models but did not state clearly in their conclusion if any can be successfully implemented to simulate air quality in the Sydney region, or what else improve in the studied models.

3.Summary and Conclusions section is too long. I suggest shortening Summary and Conclusions as it contains repetition from the Discussion.

4.Line 380: “These results are not significantly….”. Did you run significance test, if not you could use “notably”.

Author Response

(The authors gave the same response as above.)
